# Spin-Orbit induced phase-shift in $Bi_2Se_3$ Josephson junctions

Alexandre Assouline[1,2], Cheryl Feuillet-Palma[1], Nicolas Bergeal [1], Tianzhen Zhang[1], Alireza Mottaghizadeh[1,5], Alexandre Zimmers[1], Emmanuel Lhuillier[3], Mahmoud Eddrie[3], Paola Atkinson[3], Marco Aprili[4] & Hervé Aubin[1,6]

The transmission of Cooper pairs between two weakly coupled superconductors produces a superfluid current and a phase difference; the celebrated Josephson effect. Because of time-reversal and parity symmetries, there is no Josephson current without a phase difference between two superconductors. Reciprocally, when those two symmetries are broken, an anomalous supercurrent can exist in the absence of phase bias or, equivalently, an anomalous phase shift $\varphi_0$ can exist in the absence of a superfluid current. We report on the observation of an anomalous phase shift $\varphi_0$ in hybrid Josephson junctions fabricated with the topological insulator $Bi_2Se_3$ submitted to an in-plane magnetic field. This anomalous phase shift $\varphi_0$ is observed directly through measurements of the current-phase relationship in a Josephson interferometer. This result provides a direct measurement of the spin-orbit coupling strength and open new possibilities for phase-controlled Josephson devices made from materials with strong spin-orbit coupling.

[1] LPEM, ESPCI Paris, PSL Research University; CNRS; Sorbonne Universités, UPMC University of Paris 6, 10 rue Vauquelin, F-75005 Paris, France. [2] Service de Physique de l'État Condensé, CNRS UMR 3680, IRAMIS, CEA-Saclay, 91191 Gif-sur-Yvette, France. [3] Sorbonne Universités, UPMC Univ. Paris 06, CNRS-UMR 7588, Institut des NanoSciences de Paris, 4 place Jussieu, 75005 Paris, France. [4] Laboratoire de Physique des Solides, CNRS, Univ. Paris-Sud, University Paris-Saclay, 91405 Orsay Cedex, France. [5] Present address: Khatam University, 30 Hakim Azam Street, Tehran 1991633356, Iran. [6] Present address: Centre de Nanosciences et de Nanotechnologies, CNRS, Univ. Paris-Sud, Universités Paris-Saclay, C2N, 91120 Palaiseau, France. Correspondence and requests for materials should be addressed to A.A. (email: alexandre.assouline@cea.fr) or to H.A. (email: herve.aubin@c2n.upsaclay.fr)

I n Josephson junctions[1], the Current-Phase relationship (CPR) is given by the first Josephson equation[2], $I_J(\varphi) = I_0 \sin (\varphi + \varphi_0)$. Time-reversal and spatial parity symmetries[3], $P_x, P_y, P_z$ impose the equality $I_J(\varphi \rightarrow 0) = 0$ and so only two states for the phase shift $\varphi_0$ are possible. $\varphi_0 = 0$ in standard junctions and $\varphi_0 = \pi$ in presence of a large Zeeman field, as obtained in hybrid superconducting-ferromagnetic junctions[4–6] or in large g-factor materials under magnetic field[7,8].

To observe an anomalous phase shift $\varphi_0$ intermediate between 0 and $\pi$, both time-reversal and parity symmetries must be broken[9–22]. This can be obtained in systems with both a Zeeman field and a Rashba spin-orbit coupling term $H_R = \frac{\alpha}{\hbar}(\mathbf{p} \times \mathbf{e}_z).\sigma$ in the Hamiltonian[11,20], where $\alpha$ is the Rashba coefficient, $\mathbf{e}_z$ the direction of the Rashba electric field and $\sigma$ a vector of Pauli matrices describing the spin. Physically, these terms lead to a spin-induced dephasing of the superconducting wave-function.

This anomalous phase shift is related to the inverse Edelstein effect observed in metals or semiconductors with strong spin-orbit coupling. While the Edelstein effect consists in the generation of a spin polarization in response to an electric field[23], the inverse Edelstein effect[24], also called spin-galvanic effect, consists in the generation of a charge current by an out-of-equilibrium spin polarization. These two magneto-electric effects are predicted also to occur in superconductors as a consequence of a Lifshitz type term in the free energy[25,26]. Thus, in a superconductor with a strong Rashba coupling, a Zeeman field induces an additional term in the supercurrent. In Josephson junctions this term leads to the anomalous phase shift[20].

Several designs of Josephson junctions leading to an anomalous phase shift have been proposed theoretically where the Zeeman field can be obtained from an applied magnetic field[14,16,20] or by using a magnetic element[11]. These designs include the use of atomic contacts[12], quantum dots[13,27], nanowires[17,18], topological insulator[19,28–30], diffusive junction[20], magnetic impurity[21], ferromagnetic barrier[11], and diffusive superconducting-ferromagnetic junctions with non-coplanar magnetic texture[31].

Experimentally, the anomalous phase shift $\varphi_0$ can be detected in a Josephson Interferometer (JI) through measurements of the CPR. Anomalous phase shifts have been identified recently in JIs fabricated from the parallel combination of a normal '0' and 'π' junction[22] that breaks the parity symmetries.

Because of its large g-factor $g = 19.5$[32] and large Rashba coefficient, $Bi_2Se_3$ is a promising candidate for observing the anomalous Josephson effect due to the interplay of the Zeeman field and spin-orbit interaction. In this topological insulator[33,34], the effective Rashba coefficient of the topological Dirac states is about $\alpha \simeq 3$ eVÅ[35], while the Rashba coefficient of the bulk states, induced by the broken inversion symmetry at the surface, has a value in the range 0.3–1.3 eVÅ as measured by photoemission[36,37].

As detailed in refs. [20,26] the amplitude of the anomalous phase depends on the amplitude of the Rashba coefficient $\alpha$, the transparency of the interfaces, the spin relaxation terms such as the Dyakonov-Perel coefficient and whether the junction is in the ballistic or diffusive regime. At small $\alpha$, the anomalous phase is predicted proportional to $\alpha^3$, at large $\alpha$, it should be proportional to $\alpha$.

In the ballistic regime[11] and for large $\alpha$, the anomalous phase shift is given by $\varphi_0 = \frac{4E_Z \alpha L}{(\hbar v_F)^2}$ for a magnetic field of magnitude $B$ and perpendicular to the Rashba electric field, where $E_Z = \frac{1}{2}g\mu_B B$ is the Zeeman energy, $L$ is the distance between the superconductors and $v_F$ is the Fermi velocity of the barrier material. For the Rashba spin-split conduction band with $\alpha \approx 0.4$ eVÅ, $v_F = 3.2 \times 10^5$ ms$^{-1}$ and junction length $L = 150$ nm, a magnetic field $B = 100$ mT generates an anomalous phase $\varphi_0 \simeq 0.01\pi$, while for Dirac states[35] with $v_F = 4.5 \times 10^5$ ms$^{-1}$, $\varphi_0 \simeq 0.005\pi$.

In the diffusive regime, the expected anomalous phase shift has been calculated in ref. [20]. For weak $\alpha$, highly transparent interfaces and neglecting spin-relaxation, the anomalous phase shift is given by the relation:

$$\varphi_0 = \frac{\tau m^{*2} E_Z (\alpha L)^3}{3\hbar^6 D} \qquad (1)$$

where $\tau = 0.13$ ps is the elastic scattering time, $D = \frac{1}{3}v_F^2\tau = 40$ cm$^2$ s$^{-1}$ is the diffusion constant and $m^* = 0.25\, m_e$ is the effective electron mass[38].

To test these theoretical predictions, we fabricated single Josephson junctions and JIs from $Bi_2Se_3$ thin films of 20 quintuple layers thick, ~20 nm, grown by Molecular Beam Epitaxy and protected by a Se layer, see Supplementary Note 1 and Supplementary Figure 1. As described in Supplementary Figure 2, these junctions are in the diffusive regime. From the measurement of the relative phase shift between two JIs with different orientations of the Josephson junctions with respect to the in-plane magnetic field, we observed unambiguously the anomalous phase-shift predicted by Eq. (1).

## Results

**Current-phase relationship and Shapiro steps.** The JI shown in Fig. 1a consists of two junctions in parallel of widths $W_1 = 600$ nm and $W_2 = 60$ nm, respectively. The phase differences $\varphi_1$ and $\varphi_2$ for the two junctions are linked by the relation $\varphi_1 - \varphi_2 = 2\pi\frac{\phi}{\phi_0}$, where $\phi = B_z S$ is the magnetic flux enclosed in the JI of surface $S$, $B_z$ is a small magnetic field perpendicular to the sample, i.e. along $\mathbf{e}_z$, and $\phi_0$ is the flux quantum. In this situation, the Zeeman energy is negligible and oriented along the Rashba electric field, which implies that $\varphi_0 = 0$. As the critical current $I_{c_1}$ is much higher than $I_{c_2}$, then $\varphi_1 = \pi/2$ and $I_c = I_{c_1} + I_{c_2} \cos(\omega B_z)$ with $\omega = 2\pi S/\phi_0$[39]. Thus, a measurement of the critical current $I_c$ as function of $B_z$ provides a measure of the current $I_{c_2}$ as function of $\varphi_2$, i.e. the CPR. From the voltage map as a function of current $I$ and $B_z$, shown Fig. 1b, the critical current $I_c$ is extracted when the voltage across the device exceeds the value $V_{switch} = 4 \mu V$, as shown in Fig. 1c. We find that the CPR displays a conventional sinusoidal form $I_J = I_c \sin (\varphi)$, as shown by the fit in Fig. 1d. Furthermore, under microwave irradiation, the JI displays a conventional, $2\pi$ periodic, Shapiro pattern, as shown Fig. 2, and detailed discussion in Supplementary Note 2.

**Asymmetric Fraunhofer pattern with in-plane magnetic field.** Figure 3b–d show resistance maps d$V$/d$I$ of a single junction, Fig. 3a, as function of current $I$ and $B_z$ for different values of an in-plane magnetic field, $B_y$. Figure 3e shows the corresponding critical current curves. A Fraunhofer pattern is observed with the first node located at $B_0 \simeq 1.2$ mT. This value is consistent with the theoretical value $B_0 = \frac{\phi_0}{W(L+2\lambda_z)}$, using the effective magnetic penetration depth $\lambda_z = 175$ nm and taking into account flux-focusing effects, see Supplementary Note 3. While for $B_y = 0$, the Fraunhofer pattern is symmetric with respect to $B_z$, this pattern becomes asymmetric upon increasing the amplitude of $B_y$. This evolution is shown in the critical current map as a function of $B_z$ and $B_y$ in Fig. 3f. We observed a much less pronounced asymmetry when we apply the magnetic field in the $x$ direction as shown in Supplementary Figure 5. Similar behavior has been observed recently in InAs[40] interpreted as the consequence of a

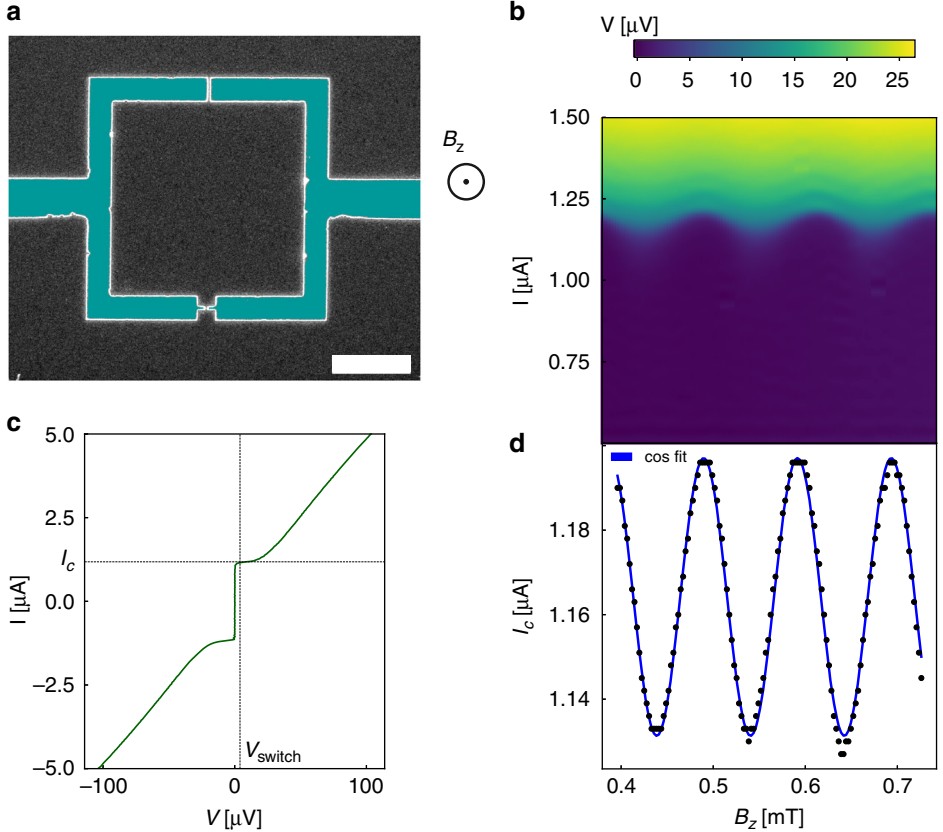

**Fig. 1** Probing the CPR with a Josephson interferometer. **a** SEM image of a JI device. The scale bar is 2 μm. This device consists of two Josephson junctions in parallel and enables probing the CPR of the smaller junction. The Aluminum electrodes are 50 nm thick. **b** Voltage map as a function of current and magnetic field $B_z$ applied perpendicular to the film plane showing zero voltage below a critical current value. The critical current of the small junction oscillates with the magnetic flux in the superconducting loop. **c** IV curve of the JI device. The critical current value $I_c$ is extracted when the junction develops a finite voltage defined as $B_z$ and indicated by the dashed vertical line. **d** Critical current extracted from **c** as a function of $B_z$. The oscillations are properly described by the function $I_c(B_z) = I_{c_1} + I_{c_2} \cos(\omega B_z)$, indicating a sinusoidal CPR as expected for conventional Josephson junctions

combination of spin-orbit, Zeeman and disorder effects. As described in ref. [3], the generation of an anomalous phase shift requires breaking all symmetry operations U leaving $UH(\varphi)U^\dagger = H(-\varphi)$, where H is the full Hamiltonian of the system including spin-orbit interactions. These symmetry operations are shown in Table 1 together with the parameters breaking those symmetries. This table shows that for a system with a finite spin-orbit coefficient $\alpha$, finite $B_y$ is sufficient to generate an anomalous phase shift. However, additional symmetry operations U leaving $UH(B_z, \varphi)U^\dagger = H(-B_z, \varphi)$ must be broken to generate an asymmetric Fraunhofer pattern, as shown in Supplementary Table 1. In addition to non-zero values for $\alpha$ and $B_y$, disorder along $y$ direction, i.e. non-zero $V_y$, is required to generate an asymmetric Fraunhofer pattern. AFM images, as in shown in Supplementary Figure 6g, show that the MBE films present atomic steps. Due to the dependence of the Rashba coefficient on film thickness[35], phase jumps along the $y$ direction of the junction can be produced by jumps in the Rashba coefficient and explains the polarity asymmetry of the Fraunhofer pattern. As detailed in Supplementary Note 4, using a simple model, the asymmetric Fraunhofer pattern measured experimentally can be simulated, as shown in Fig. 3g.

**Current-phase relationship with in-plane magnetic field**. To unambiguously demonstrate that an anomalous phase shift $\varphi_0$ can be generated by finite spin-orbit coefficient $\alpha$ and finite

magnetic field $B_y$ alone, a direct measurement of the CPR with in-plane magnetic field is required. To that end, we measured simultaneously two JIs, oriented as sketched in Fig. 4a, differing only by the orientation of the small junctions with respect to the in-plane magnetic field.

The CPRs for the two JIs are measured as function of a magnetic field making a small tilt $\theta$ with the sample plane, which produces an in-plane $B_y = B \cos(\theta)$ and a perpendicular $B_z = B \sin(\theta)$ magnetic field, as sketched in Fig. 4c. In this situation, the critical current for the reference JI changes as $I_c \propto \cos(\omega_{ref} B)$ with $\omega_{ref} = \frac{2\pi S_{ref}}{\phi_0} \sin(\theta)$ where $S_{ref}$ is the surface of the JI. For the anomalous JI, the critical current changes as $I_c \propto \cos(\omega B)$ with:

$$\omega = \frac{2\pi S}{\phi_0} \sin\theta + C_{\varphi_0} \cos(\theta) \qquad (2)$$

where $C_{\varphi_0} = \frac{\tau m^{*2} g \mu_B (\alpha L)^3}{6 \hbar^6 D}$ in the diffusive regime.

In Eq. (2), the first term arises from the flux within the JI of area $S$, the second term arises from the anomalous phase shift $\varphi_0 = C_{\varphi_0} B \cos(\theta)$.

Figure 4b shows voltage maps for two different orientations $\theta$, Fig. 4c. At low $B$, the two JIs are in-phase and become out-of-phase at higher magnetic field, indicating that the frequency $\omega$ of the anomalous JI is slightly larger than the reference JI. This is also visible on the critical current plot, Fig. 5a, extracted from these voltage maps. To see this more clearly, the average critical current, shown as a continuous line in Fig. 5a, is removed from

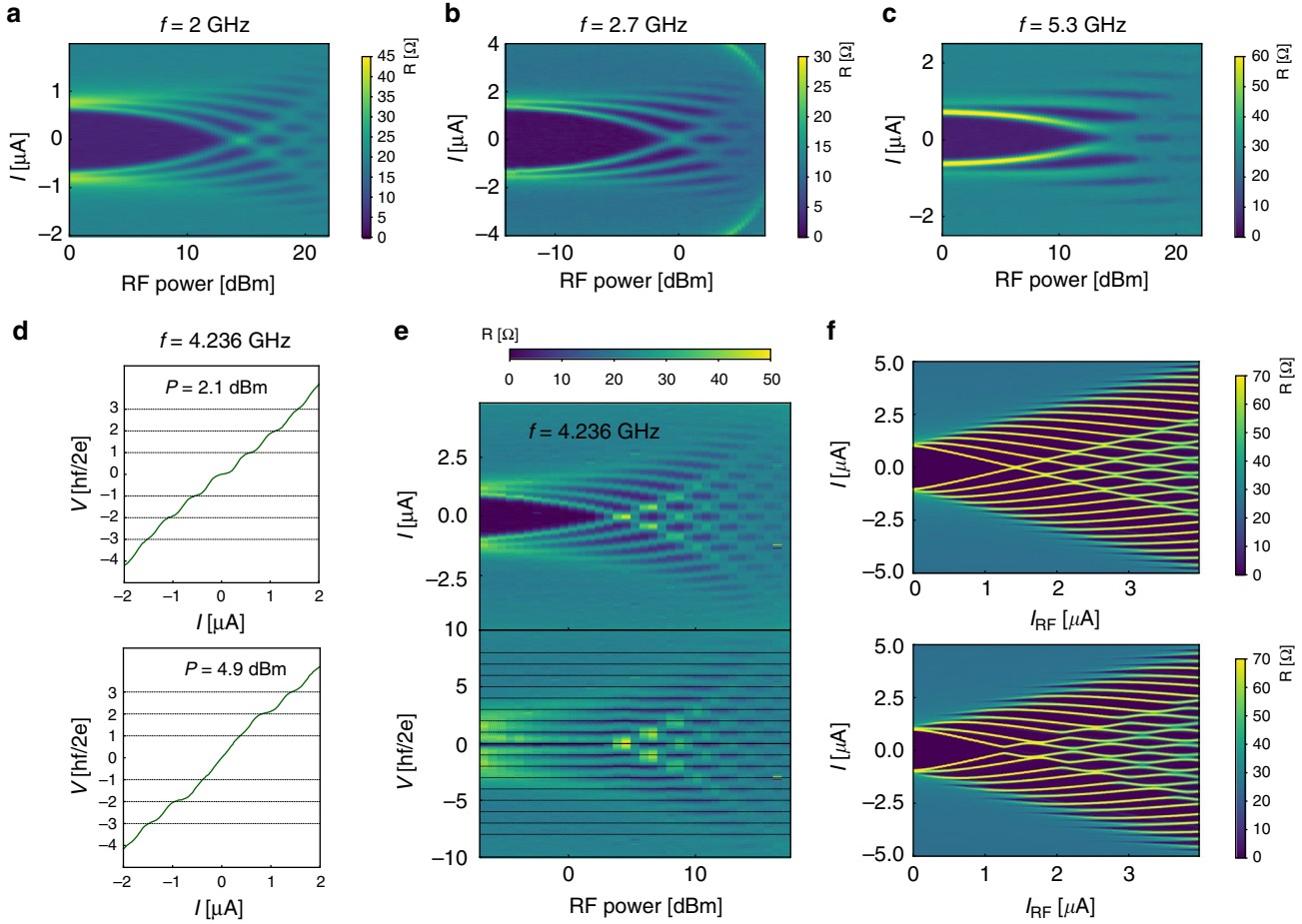

**Fig. 2** A.C. Josephson effect in $Bi_2Se_3$. For this Josephson junction, 50 nm thick Aluminum electrodes are used. **a–c** Resistance maps as a function of current and RF power for different microwave frequencies (a) $f = 2$ GHz, (b) 2.7 GHz, and (c) 5.3 GHz. The zero resistance regions correspond to voltage plateaus. **d** IV curves showing the Shapiro steps for two values of the microwave power. The $n^{th}$ current step appears at a voltage $V_n = \frac{nhf}{2e}$. **e** Resistance map as a function of current (upper panel) or voltage (lower panel) and RF power at the frequency $f = 4.236$ GHz. In the lower panel as a function of voltage, dashed lines are plotted at $V_n = \frac{nhf}{2e}$. **f** Theoretical maps calculated at $f = 4.236$ GHz with the RSJ model, see Supplementary Note 2, using two different CPRs: a conventional $2\pi$ periodic CPR $I_J = I_c\sin(\varphi)$ is used for upper panel, it reproduces properly the experimental data. An unconventional CPR $I_J = I_c\left(\frac{4}{5}\sin(\varphi) + \frac{1}{5}\sin(\varphi/2)\right)$ is used for the lower panel where the odd steps have a lower current amplitude

the critical current curve and the result shown in Fig. 5b for the two JIs. On these curves, the nodes at $\pi(2n + 1/2)$, $n = 0,1,..$, are indicated by large red (blue) dots for the reference (anomalous) curve. At low magnetic field, the two JIs are in-phase as indicated by the blue and red dots being located at the same field position. Upon increasing the in-plane magnetic field, the two JIs become out-of-phase with the anomalous JI oscillating at a higher frequency than the reference JI, as indicated by the blue dot shifting to lower magnetic field position with respect to the red dot. This increased frequency for the anomalous device is expected from Eq. (2) as a consequence of the anomalous phase shift. Supplementary Figure 7a,b shows additional data taken from negative to positive magnetic field, across zero magnetic field. A plot of the phase difference between the two JIs as function of in-plane magnetic field, shown in Supplementary Figure 7c, demonstrates that the two JIs are in-phase at zero magnetic field and reach a dephasing approaching about $\pi/2$ for an in-plane magnetic field of $\simeq 80$ mT.

One also sees that the oscillation period of both JIs increase with increasing $B$. As detailed in Supplementary Note 3, this is due to flux focusing that makes the effective area of the JIs larger at low magnetic field. As the effect of flux focusing decreases with the increasing penetration depth at higher magnetic field, the

effective areas of the JIs decreases upon increasing the magnitude of the magnetic field and so the period of oscillations increases.

While the two JIs have been fabricated with nominally identical areas, to exclude that the observed difference in frequencies between the two JIs is due to a difference of areas, we plot in Fig. 5c, the frequency ratio $\frac{\omega}{\omega_{ref}}(\theta)$ measured at different angles $\theta$. Because each curve contains several periods $T_i$, the frequency ratio is obtained from the average between the $N$ periods ratio as $\omega/\omega_{ref} = \frac{1}{N}\sum_{i=1}^{N}\frac{T_{i,ref}}{T_i}$, where $T_{i,ref}$ and $T_i$ are the $i^{th}$ oscillation period for the reference and for the anomalous device respectively. This method enables ignoring the flux focusing effect because the ratio is only taken between two periods measured at about the same magnetic field. We find that the experimental data follows the relation:

$$\frac{\omega}{\omega_{ref}}(\theta) = \frac{S}{S_{ref}} + \frac{C_{\varphi_0}\phi_0}{2\pi S_{ref}\tan(\theta)} \qquad (3)$$

At large $\theta$, this ratio is equal to the ratio of areas $S/S_{ref} \simeq 1$, however, for small $\theta$, this ratio increases as $1/\tan(\theta)$, indicating the presence of an anomalous phase shift $\varphi_0$.

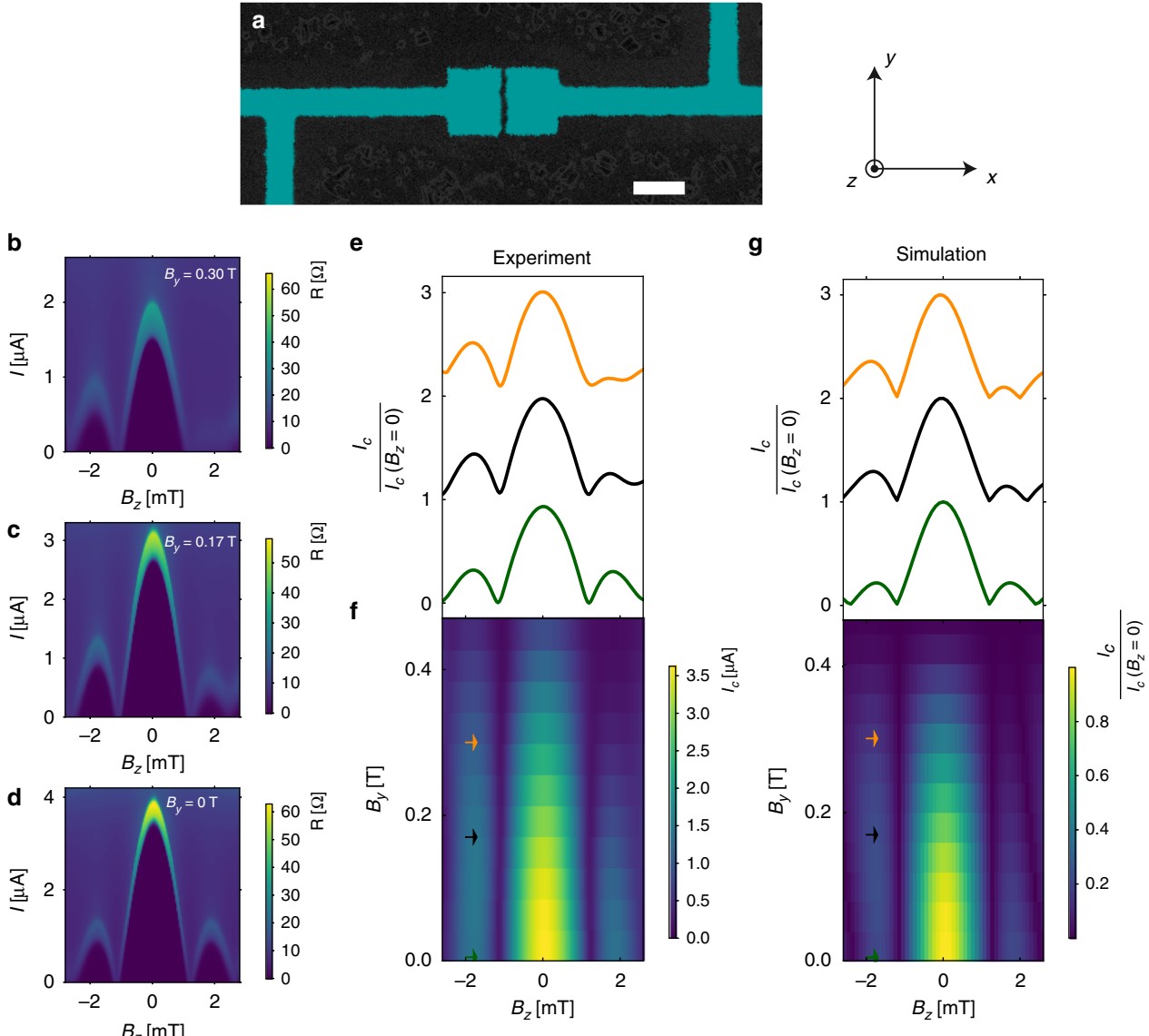

**Fig. 3** Asymmetric Fraunhofer pattern. **a** SEM image of a single junction device with 20-nm-thick Aluminum electrodes. The scale bar is 2 μm. The spacing between the two superconducting electrodes is $L = 150$ nm and the width of the junction $W = 2$ μm. **b–d** Resistance maps as function of current and perpendicular magnetic field $B_z$, for different values of the in-plane magnetic field $B_y$. The corresponding critical current curves are shown panel **e**. At zero in-plane magnetic field, panel **d**, the Fraunhofer pattern is symmetric with respect to $B_z$. The pattern becomes asymmetric upon increasing $B_y$. **f** Critical current map as a function of $B_z$ and $B_y$ showing the evolution from symmetric Fraunhofer pattern at $B_y = 0$ to the asymmetric Fraunhofer pattern for $B_y > 0$. **g** Numerical simulation of the anomalous Fraunhofer pattern due to disorder and spin-orbit coupling. The model is described in the Supplementary Note 4

| UH($\varphi$)U$^{\dagger}$ = H($-\varphi$) | Broken by |
|---|---|
| $P_y P_x$ | $\alpha$, $V_x$, $V_y$ |
| $\sigma_z P_y P_x$ | $B_x$, $B$, $V_x$, $V_y$ |
| $\sigma_x P_y T$ | $B_x$, $\alpha$, $V_y$ |
| $\sigma_y P_y T$ | $B_y$, $V_y$ |

**Table 1 Symmetry operations U protecting H($\varphi$) = H($-\varphi$) from ref. [3]**

The symmetry operations on the left column are broken by one of the parameters on the right column. These parameters include the in-plane magnetic fields $B_x$, $B_y$, the asymmetric disorder potentials $V_x$, $V_y$, and the spin-orbit term $\alpha$ which is the consequence of the structural inversion asymmetry (Rashba). To generate an anomalous phase $\varphi_0$, each symmetry operator, one per line of the table, must be broken. For example, the combination of the magnetic field $B_y$ and the spin-orbit coupling $\alpha$ is enough to break all symmetries.

Another way of extracting the frequency is described in the Supplementary Note 5 and leads to the same result, as shown in Supplementary Figure 8.

## Discussion

A fit of the experimental data with Eq. (3), Fig. 5c, enables extracting the coefficient $\frac{C_{\varphi_0}\phi_0}{2\pi S_{ref}} = 41 \pm 510^{-5}$. Using the expression of $C_{\varphi_0}$ in the diffusive regime given above, we calculate a spin-orbit coefficient $\alpha = 0.38 \pm 0.015$ eVÅ. This value of the Rashba coefficient is consistent with the value extracted from Rashba-split conduction band observed by photoemission measurements[36,37]. Table 2 gives the anomalous phase shift extracted from the critical current oscillations at the largest magnetic field about 80–100 mT. The phase shift is extracted from the magnetic

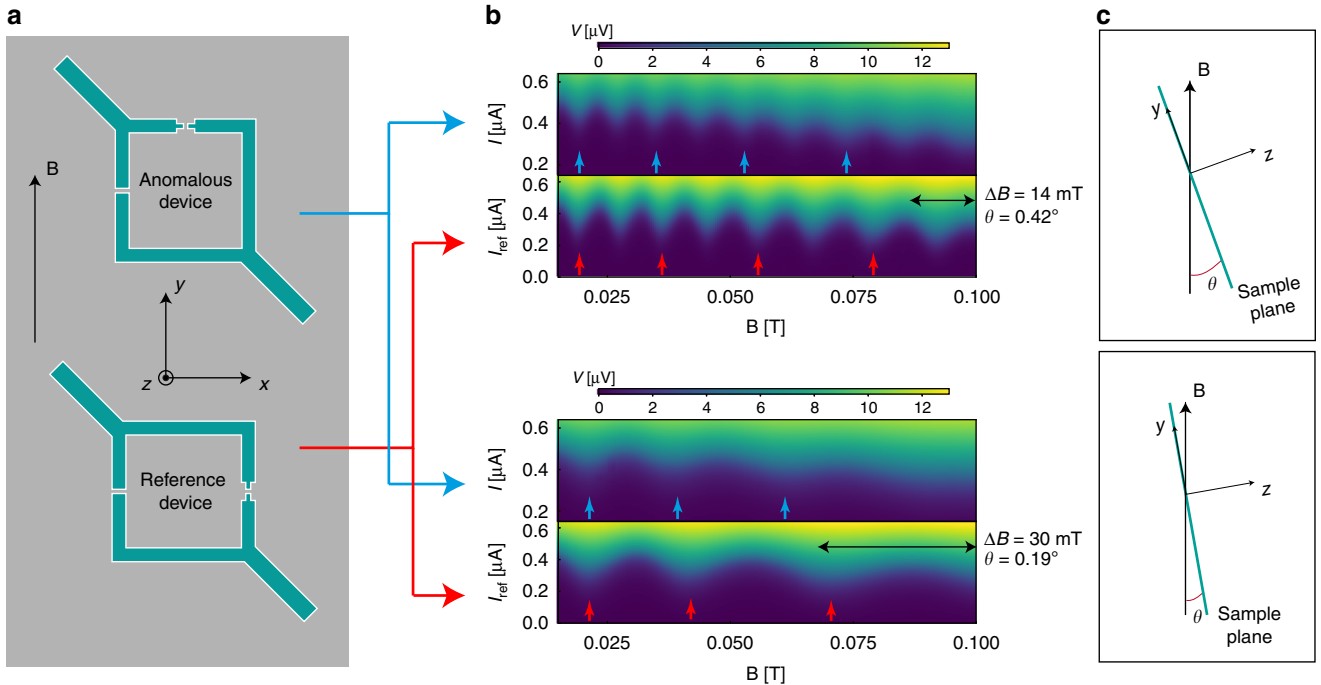

**Fig. 4** Probing the anomalous phase shift. **a** Sketch of the setup consisting of two JIs fabricated on the same chip. The reference and anomalous JIs have identical area, respectively $S$ and $S_{ref}$, where $S \simeq S_{ref} \simeq 20.6\,\mu m^2$. The Aluminum electrodes are 20 nm thick. From Table 1, in the absence of disorder, an anomalous phase shift induced by Rashba spin-orbit coupling can be generated only by an in-plane magnetic field $B_y$. **b** Voltage map showing the critical current oscillation of the two devices as a function of magnetic field B. The critical current of both devices oscillate due to the perpendicular component of the magnetic field $B_z = B \sin(\theta)$, as sketched panel **c**. The oscillation frequency can be changed by mechanically tilting the sample, i.e by changing the angle $\theta$ between the plane containing the superconducting loop and the magnetic field B. The frequency of the anomalous device is larger than the reference as a consequence of the anomalous phase shift. The colored arrows are guide to the eyes, to help visualizing the increased phase shift in the anomalous device

**Table 2 Anomalous phase shifts obtained at $B \simeq 100$ mT, compared to theory**

|  | $\theta = 0.1°$ | $\theta = 0.22°$ | $\theta = 0.46°$ | Ballistic | Dirac | Diffusif |
|---|---|---|---|---|---|---|
| $\varphi_0$ | $0.88\pi$ | $1.01\pi$ | $0.85\pi$ | $0.01\pi$ | $0.005\pi$ | $0.94\pi$ |

The three first columns show the anomalous phase shifts extracted from the last nodes of the critical current oscillations shown in Fig. 5 for the three curves taken at different angles $\theta$. The last three columns show the calculated anomalous phase shifts in the ballistic regime, for the Rashba-split conduction states and Dirac states, and in the diffusive regime, for the Rashba-split conduction states. These theoretical values are calculated using $\alpha = 0.4$ eVÅ for the Rashba-split conduction states and $\alpha = 3$ eVÅ for the Dirac states. See main text for the formula and other parameters.

field difference between the last nodes of the oscillations, indicated by blue and red dots on Fig. 5. At this largest magnetic field, we find an anomalous phase shift $\varphi_0 \simeq 0.9\pi$ for all three tilt angles $\theta$. This shows that the anomalous phase shift depends only on the parallel component of the magnetic field as expected. This experimental value is compared with the theoretical values calculated in the ballistic regime, for the Rashba-split conduction states and Dirac states, and in the diffusive regime, for the Rashba-split conduction states. This table shows that the Dirac states provide only a phase shift of $0.005\pi$ and so cannot explain the experimental data. The table also shows that the Rashba-split conduction states provide a phase shift of only $0.01\pi$ in the ballistic regime while they provide an anomalous shift of $0.94\pi$ in the diffusive regime, close to the experimental value, confirming that the junctions are indeed in the diffusive regime and demonstrating the validity of the theory leading to Eq. (1).

A detailed look at Table 1 shows that the anomalous shift observed here must be the consequence of finite Rashba coefficient and in-plane magnetic field. While Table 1 shows that disorder alone $V_y$ is sufficient to generate an anomalous phase shift, this disorder-induced anomalous phase shift should exist even at zero magnetic field and should not change with magnetic field. In contrast, as discussed above, we have seen that the two JIs are in-phase near zero magnetic field and become out of phase only for at finite magnetic field. Thus, this observation implies that disorder $V_y$ is absent, which is plausible as the small Josephson junction is only 150 nm × 150 nm. In the absence of disorder $V_y$, Table 1 shows that the only way for an anomalous phase shift to be present is that the coefficient $\alpha$ be non-zero. Indeed, if $\alpha$ were zero, the first and third symmetry operations of Table 1 would not be broken even with finite $B_y$.

To summarize the result of this work, the simultaneous measurements of the CPR in two JIs making an angle of 90° with respect to the in-plane magnetic field enabled the identification of the anomalous phase shift $\varphi_0$ induced by the combination of the strong spin-orbit coupling in $Bi_2Se_3$ and Zeeman field. This anomalous phase shift can be employed to fabricate a phase battery, a quantum device of intense interest for the design and fabrication of superconducting quantum circuits[41,42].

## Methods

**Sample preparation.** The $Bi_2Se_3$ samples were grown by Molecular Beam Epitaxy. The crystalline quality of the films was monitored in-situ by reflection high energy electron diffraction and ex situ by x-ray diffraction, and by post growth verification of the electronic structure though the observation of the Dirac cone fingerprint in angle-resolved photoemission spectra as described in ref. [43]. Following growth, the samples were capped with a Se protective layer. The Josephson junctions are

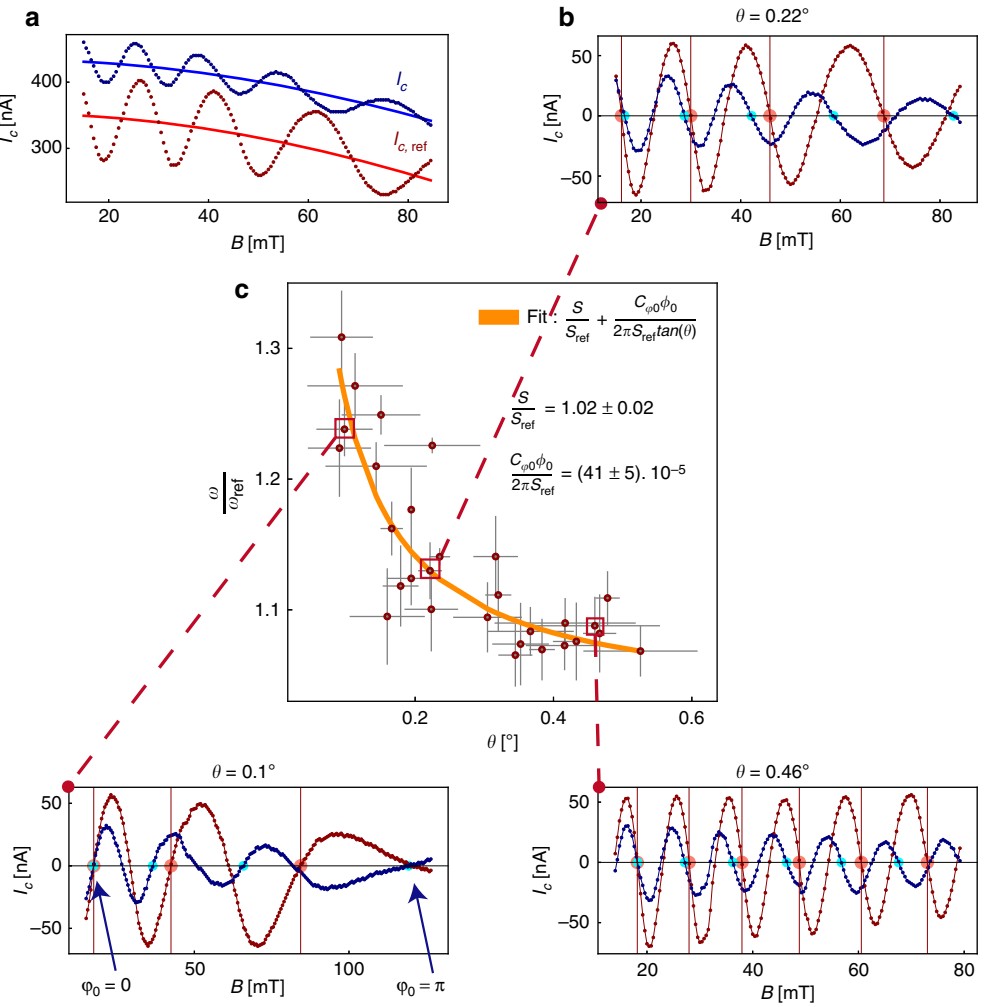

**Fig. 5** Comparison of the JIs frequencies as a function of the angle $\theta$. **a** The critical current is extracted and shown as a function of magnetic field. The red and blue curves correspond to the reference and anomalous device, respectively. As the critical current of the large junction decreases with the magnetic field, this leads to a decreasing background fitted for both devices and shown by the continuous lines. **b** Critical current as a function of magnetic field where the background, indicated by continuous lines in panel **a**, are subtracted. The anomalous device shows a larger oscillation frequency than the reference device. The colored dots help to visualize the increased frequency of the anomalous device (blue dots) with respect to the reference device (red dots). The two JIs are in-phase at low magnetic field and become out-of-phase at high magnetic field, with an anomalous phase shift reaching $\varphi_0 \simeq \pi$ for $B \simeq 100$ mT. The periods are compared one by one and the average of the single period ratios provides the value $\frac{\omega}{\omega_{\mathrm{ref}}}$. **c** The ratio of the oscillation frequencies is plotted as a function of the angle $\theta$. Without the generation of an anomalous phase, this ratio should be constant and equal to the surface ratio $\frac{S}{S_{\mathrm{ref}}} \simeq 1$. This ratio diverges as $\frac{1}{\theta}$ for small $\theta$, as shown by Eq. (3). Fitting the curve with Eq. (3) provides the spin-orbit coefficient $\alpha$

fabricated on these thin films with standard e-beam lithography, e-beam deposition of Ti(5 nm)/Al(20–50 nm) electrodes and lift-off. The Se capping layer is removed just before metal evaporation by dipping the samples in a NMF solution of Na$_2$S. In the evaporator chamber, the surface is cleaned by moderate in-situ ion beam cleaning of the film surface before metal deposition. While for standard junctions, an aluminum layer 50 nm thick is usually deposited, we also fabricated junctions with 20-nm-thick electrodes to increase their upper critical field as required by the experiments with in-plane magnetic field. See Supplementary Figure 3 for a lateral sketch of the devices. After microfabrication, the carrier concentration is about $10^{19}$ cm$^{-3}$ and the resistivity about 0.61 m$\Omega$.cm, as shown in Supplementary Figure 2. A comparison between the normal state junction resistance of the order of 20–50 $\Omega$ and the resistivity of the films indicates negligible contact resistance, i.e. the junction resistance is due to the Bi$_2$Se$_3$ film between the electrodes.

**Measurements details**. The values for the normal resistance and critical current values measured on 20 devices are found to be highly reproducible, demonstrating the reliability of our procedure for surface protection and preparation before evaporation of the electrodes. The devices are measured in a dilution fridge with a base temperature of 25 mK. The IV curves are measured with standard current source and low noise instrumentation amplifiers for detecting the voltage across the junctions. The measurement lines are heavily filtered with $\pi$ filters at room

temperature at the input connections of the cryostat. They are also filtered on the sample stage at low temperature with 1 nF capacitances connecting the measurements lines to the ground.

## Data availability

The data that support the main findings of this study are available from the corresponding author upon request. The source data underlying Fig. 5c and Supplementary Fig. 8d are provided as Source Data files.

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

## Acknowledgements
The devices have been made within the consortium Salle Blanche Paris Centre. We acknowledge fruitful discussions with S. Bergeret and J. Danon. This work was supported by French state funds managed by the ANR within the Investissements d'Avenir programme under reference ANR-11-IDEX-0004-02, and more specifically within the framework of the Cluster of Excellence MATISSE. We also thanks L. Largeau (C2N: Centre de Nanosciences et de Nanotechnologies-Universit Paris-Sud) and D. Demaille (INSP: Institut des NanoSciences de Paris-Sorbonne Univerit)) for the atomic-resolved HAADF-STEM images.

## Author contributions
H.A. proposed and supervised the project. M.E. and P.A. have grown the Bi₂Se₃ thin films by MBE and did the structural characterization (AFM, X-Ray, and TEM). A.A. designed and microfabricated the samples with the help of C.F.P., T.Z., A.M., A.Z., E.L., M.A., and H.A.; A.A., M.A., and H.A made the measurements with the help of C.F.P. and N.B.; A.A., M.A., and H.A. analyzed the data and wrote the manuscript.

## Additional information

**Competing interests:** The authors declare no competing interests.

