## [Peer Review File · Nature Communications]

Reviewers' comments:

Reviewer #1 (Remarks to the Author):

In this manuscript the authors present an interesting experiment that shows the realisation of Josephson junctions with a non-trivial current-phase relation (CPR). Specifically, they observed a phase shift in the CPR which can be related to the interplay of spin-orbit interaction (SOI) and Zeeman field. Although such a shift had been observed in a previous experiment (ref. 24) on quantum dots, in the present manuscript the observation is done in a completely different material, BiSe, and in my opinion the analysis is more careful. Overall the article is well written and I will recommend the article for publication in Nat. Comm. However, I would recommend some fine-tuning of the introduction, discussion of results and references.

In the introduction the authors discussed in terms of symmetry the appearance of the anomalous phase (AP). This is definitely a good way to start, however I recommend to be more general. In particular in the second paragraph the authors immediately jump to the discussion of the anomalous phase (AP) due to Rashba term. In principle there is no proof that the SOI in their junctions are described by Rashba-like AP. AP may appear for more general SOI as demonstrated in Konshelle et al, PHYSICAL REVIEW B 92, 125443 (2015). Moreover, as clearly explained in Silaev et al, PHYSICAL REVIEW B 95, 184508 (2017), AP may also appear in the case of a non-coplanar magnetic texture. Therefore I would recommend in the first paragraphs to include more generic statements about the appearance of the AP.

It is clear that in the present experiment the AP is due to the SOI in BiSe and therefore one can mention this after this general discussion. However, and as mentioned above, it is not clear that the Rashba-like describes the SOI in BiSe. This might be a good approximation to give some numbers later on in the experimental analysis of Figs. 3-4. As demonstrated by Konshelle et al, PHYSICAL REVIEW B 92, 125443 (2015), the AP is a more general effect directly connected to the spin-galvanic effect in systems with SOI. A connection between the AP and the spin-galvanic effect in the introduction will definitely broaden the scope of the article and make the paper also attractive for scientists working on spintronics.

Regarding the explanation of the experimental data, several assumptions have been done that probably have to be clarified. In my opinion the discussion in pages 3-4 on ballistic vs diffusive is not accurate and may lead to misunderstandings. As demonstrated by Konshelle et al, PHYSICAL REVIEW B 92, 125443 (2015) and in Ref. 22, the linear or cubic dependence on α^*L is not characteristic to the ballistic or diffusive case but depends on the strength of the SOI, interface quality, etc. I am not requiring to analyse this in detail since it is probably not easy to determine all the theory parameters from the experimental data. I suppose that the authors tried different fittings with different equations that they found in the literature and used the one fitting at best. And this is a completely valid procedure. However, I would recommend to emphasise this and when using for example Eqs. (1-2), by fitting the data, to mention the assumptions made. For example, as far as I understand from the papers mentioned above Eq. 1 is valid in the case of weak α and after neglecting the spin-relaxation terms.

Review of Spin-Orbit induced phase-shift in Bi₂Se₃ Josephson junctions

In the manuscript “Spin-orbit induced phase-shift in Bi₂Se₃ Josephson junctions”, authors Assouline *et al.*, investigate the possibility of a modification of the current-phase relation to allow for an anomalous phase φ . The authors initiate the study by investigating the properties of Josephson junctions (JJs) fabricated from MBE grown thin films of Bi₂Se₃, including measuring the current-phase relation and the AC Josephson effect, and observe that a conventional sinusoidal current-phase relations occurs in these junctions. Anomalous behavior arising from strong spin-orbit coupling in Bi₂Se₃ is measured when a parallel magnetic field is applied to the JJ. This includes both an unconventional Fraunhofer pattern measured in a single JJ and a difference of oscillations frequency in dual JJ devices called “Josephson Interferometers”. An investigation of the role of spin-orbit coupling in JJs is timely, as this remains an important ingredient in both topological superconductivity and novel device behavior of JJs. However, there are remaining outstanding questions about the work that need to address before this can be published in Nature Communications.

The main questions to be answered concern Figs 4 and 5 — i.e. the data generated by the two Josephson Interferometers.

i. It is apparent that the oscillation period depends on the magnitude of the applied magnetic field. The authors state that flux focusing is the origin of the aperiodic oscillation of I_c , yet it is unclear on how this actually works or why it depends on the magnitude of the parallel field. For flux focusing to have this effect, a “flux-focused” modification of the perpendicular magnetic field would need to be present. Yet, when the Fraunhofer pattern is measured in parallel field, no variation of the perpendicular field is apparent. Can the authors address where the flux focusing is happening and why it is different in two interferometers?

Further — and perhaps as a possible alternate suggestion to why the data is not periodic — in a single interferometer, the large junction’s critical current will be affected by the perpendicular magnetic field more strongly than the small junction. In Fig. 4b for example, the perpendicular field is around 0.75mT. Is it still true that $I_c^{large} \gg I_c^{small}$?

ii. In addition, the variation of I_c is also decreased by the increasing magnetic field. One may also attribute this to flux focusing. For example, in Ref. 34 of the paper observe a “dephasing” effect from a parallel field, yet found that this was strongest when the field was applied in the “x-direction”. Therefore, it would be expected that this dephasing effect would be stronger in the reference interferometer, which is the opposite observed. Can the authors comment on the origin of this apparent dephasing effect and its stronger response to parallel magnetic field in the “Anomalous device”?

To help address points i and ii, it would be helpful if a full plot of their data from $B=0$ to $B=0.1\text{T}$ is included in the supplementary information.

iii. It is not clear why the reference device is devoid of any anomalous behavior. The authors state that the two interferometers are in sync at $B=0$ but, due to the nature of the measurement, it is not possible to directly measure that the two are in sync at $B=0$. It is simply inferred from the lowest field at which they show data. Further, the authors state that disorder V_Y is absent in the smaller devices, therefore it is possible that the reference device is just that, a device with a sinusoidal current-phase relation and no anomalous value for φ . However, looking that the AFM of Fig. S5g, it seems possible that variations of the device parameter are possible on the length scale of 150nm. Therefore, it seems that a convincing argument for the preservations of the important symmetries in the reference device is lacking.

iv. The error in extraction of the loop area is large, about two orders of magnitude larger than the value extracted for $C\varphi$. It seems possible that this error may be uncoupled to the extraction of $C\varphi$, since in the plot of $\omega/\omega_{ref}(\theta)$, only the $C\varphi$ term is dependent on theta. Yet, I could still see the error in the value of S/S_{ref} coupling into the θ -dependence of this curve. Can the authors comment on how the errors S/S_{ref} modify the errors in extracting $C\varphi$?

The above questions should be answered before publications in any journal. If the authors can answer the above however, I will strongly recommend publication in Nature Communications.

Finally, I would like to suggest some minor modifications to be made before publication.

1.) How thick is the Bi_2Se_3 thin films used in this study? The authors should be more clear about where the thin film actually is in the device. I know they attempt this in Fig. S3, but I was still confused about its location.

2.) What are the areas of the two interferometer? Without this it is not possible for the reader to calculate the extracted anomalous phase from the fit in the plot of $\omega/\omega_{ref}(\theta)$. Further, can the authors explicitly state the magnitude of the anomalous phase they extract? How does this compare to the anomalous phase predicted for ballistic, diffusive and Dirac systems?

3.) Which of the devices have 50nm of Al and which have 20nm of Al (the paper states that both are used)? This would be important in understanding the inverse proximity effect, i.e. the effect of the thin Bi_2Se_3 film on the normal state properties of the superconductor (c.f. G. Bergmann PRB 72 134505 (2005)).

4.) In the supplementary info, the authors use a sinusoidal variation of the parameter α . Does the functional form of the variation of α have a significant effect on the resulting Fraunhofer pattern?

In red: questions of the referees

In green: text added to the manuscript

Answers to Reviewer #1:

We thank the referee for his careful reading and positive evaluation of this work. We modified the manuscript following his advices.

We added a paragraph to explicit the relationship between the anomalous phase and the inverse Edelstein effect.

This anomalous phase shift is related to the inverse Edelstein effect observed in metals or semiconductors with strong spin-orbit coupling. While the Edelstein effect consists in the generation of a spin polarization in response to an electric field\cite{Edelstein1990-yj}, the inverse Edelstein effect\cite{Shen2014-lz}, also called spin-galvanic effect, consists in the generation of a charge current by a steady spin polarization. These two magneto-electric effects are predicted also to occur in superconductors as a consequence of a Lifshitz type term in the free energy\cite{Yip2002-ua,Konschelle2015-ip}. Thus, in a superconductor with a strong Rashba coupling, a Zeeman field induces an additional term in the supercurrent. In Josephson junctions this term leads to the anomalous phase shift\cite{Bergeret2015-so}.

We added the reference to Silaev2017 at the end of third paragraph.

Regarding the explanation of the experimental data, several assumptions have been done that probably have to clarify. In my opinion the discussion in pages 3-4 on ballistic vs diffusive is not accurate and may lead to misunderstandings. As demonstrated by Konshelle et al, PHYSICAL REVIEW B 92, 125443 (2015) and in Ref. 22, the linear or cubic dependence on α^*L is not characteristic to the ballistic or diffusive case but depends on the strength of the SOI, interface quality, etc. I am not requiring to analyse this in detail since it is probably not easy to determine all the theory parameters from the experimental data. I suppose that the authors tried different fittings with different equations that they found in the literature and used the one fitting at best. And this is a completely valid procedure. However, I would recommend to emphasize this and when using for example Eqs. (1-2), by fitting the data, to mention the assumptions made. For example, as far as I understand from the papers mentioned above Eq. 1 is valid in the case of weak α and after neglecting the spin-relaxation terms.

We added the ref to Konshelle et al, PHYSICAL REVIEW B 92, 125443 (2015) to precise the various parameters on which depends the anomalous phase shift and added the paragraph:

As detailed in Refs.\cite{Konschelle2015-ip,Bergeret2015-so}, the amplitude of the anomalous phase depends on the amplitude of the Rashba coefficient α , the transparency of the interfaces, the spin relaxation terms such as the Dyakonov-Perel coefficient and whether the junction is in the ballistic

or diffusive regime. At small α , the anomalous phase is predicted proportional to α^3 , at large α , it should be proportional to α .

Before Eq.1, we modified the paragraph below for an improved precision:

In the ballistic regime\cite{Buzdin2008-hg} and for large α , the anomalous phase shift is given by $\varphi_0 = \frac{4E_Z \alpha L}{\hbar v_F^2}$ for a magnetic field B perpendicular to the Rashba electric field, where $E_Z = \frac{1}{2}g\mu_B B$ is the Zeeman energy, L is the distance between the superconductors and v_F is the Fermi velocity of the barrier material. For the Rashba-split bulk conduction states with $\alpha \approx 0.4 \text{ eV \AA}$, $v_F = 3.2 \times 10^5 \text{ ms}^{-1}$ and junction length $L = 150 \text{ nm}$, a magnetic field $B = 100 \text{ mT}$ generates an anomalous phase $\varphi_0 \approx 0.01 \pi$, while for Dirac states\cite{Zhang2010-kf} with $v_F = 4.5 \times 10^5 \text{ ms}^{-1}$, $\varphi_0 \approx 0.005 \pi$.

Answers to Reviewer #2:

We thank the referee for his careful reading of the manuscript. Below, we provide answers to all the questions raised.

i. It is apparent that the oscillation period depends on the magnitude of the applied magnetic field. The authors state that flux focusing is the origin of the aperiodic oscillation of I_c , yet it is unclear on how this actually works or why it depends on the magnitude of the parallel field. For flux focusing to have this effect, a “flux-focused” modification of the perpendicular magnetic field would need be present. Yet, when the Fraunhofer pattern is measured in parallel field, no variation of the perpendicular field is apparent. Can the authors address where the flux focusing is happening and why it is different in two interferometers?

Because the thickness of the electrodes are smaller than the penetration depth, we don't expect an effect of the parallel magnetic field. However, in all these measurements of the CPR, there exists a perpendicular component $B \sin(\theta)$ of the magnetic field. This perpendicular magnetic field, with the combination of flux-focusing and vortex penetration is responsible for the increase in the oscillation period. This flux-focusing effect is identical in both interferometers.

As described in Supp.Note3, flux-focusing is responsible for an increase of the effective area of the JI at low magnetic field because of the deflection of flux lines toward the inner part of the JI. Upon increasing the magnetic field, vortices penetrate into the superconducting electrodes, reducing the density of flux lines being diverted and so reducing the effect of flux-focusing. As the effect of flux-focusing decreases, the effective area S of the JI decreases and so the oscillation period, given by $T_B = \varphi_0 / (S)$, increases.

This effect is also responsible for the increase of the period in Fraunhofer patterns, See. Fig.S5 of our supplementary, and Fig.1c of Suominen2017, (InAs based Josephson junctions) where an increase of period is also visible and explained by the penetration of the vortices.

Thus, our understanding of the increasing period is identical to that of Suominen2017. There is a small difference on the origin of vortices. In Suominen2017, they explained that Aluminum became a type II superconductor because their aluminium electrodes are very thin(10 nm). In our case, as detailed in Supp. Note 3, standard estimation of the correlations length and penetration depth suggest that 20 nm thick Al should still be a type I superconductor. We suggest, however, that the hybrid system Al on semiconductor can become a type II superconductor because of the small carrier density in Bi2Se3.

To describe the flux-focusing effect on the junction, we added the paragraph below in the main text; we also developed the Supp. Note 3 and added an additional figure Fig.S4.

One also sees that the oscillation period of both JIs decreases with increasing B . As detailed in Supp. Note 3, this is due to flux focusing that makes the effective area of the JIs larger at low magnetic field. As the effect of flux focusing decreases upon increasing the perpendicular magnetic field B_z , the effective areas of the JIs decreases upon increasing B_z and so the period of oscillations increases.

Further — and perhaps as a possible alternate suggestion to why the data is not periodic — in a single interferometer, the large junction's critical current will be affected by the perpendicular magnetic field more strongly than the small junction. In Fig. 4b for example, the perpendicular field is around 0.75mT. Is it still true that $I_c(\text{large}) \gg I_c(\text{small})$?

Yes, in all these measurements $I_c(\text{large}) \gg I_c(\text{small})$. This can be observed clearly from Fig.4 and Fig.5 where one can see that the amplitude of the oscillation of critical current (due to the small junction) is always smaller than the average critical current (given by the large junction). We restrict the analysis in a magnetic field range where: $I_c(\text{oscillation}) < I_c(\text{average})/5$.

At the angle $\theta=0.2$, the nodes of the Fraunhofer pattern for the large junction and small junction are expected at a magnetic field of $B=1.3\text{T}$ and $B=7.9\text{T}$ respectively, which are much larger than the field values where the data are taken.

ii. In addition, the variation of IC is also decreased by the increasing magnetic field. One may also attribute this to flux focusing. For example, in Ref. 34 of the paper observe a “dephasing” effect from a parallel field, yet found that this was strongest when the field was applied in the “x-direction”. Therefore, it would expected that this dephasing effect would be stronger in the reference interferometer, which is the opposite observed. Can the authors comment on the origin of this apparent dephasing effect and its stronger response to parallel magnetic field in the “Anomalous device”?

Indeed the small critical current in the anomalous device decreases faster than the small critical current of the reference, which is the opposite than observed in Ref. 34. There is a simple explanation for this effect by considering the Fraunhofer pattern induced by the flux in the xz plane of the small junction of the anomalous device. A magnetic field applied along the y-direction induces a dephasing between the two electrodes and decreases the critical current. In contrast, a magnetic field along the x-direction cannot produce a dephasing and is not expected to reduce the critical current. We estimate that for a magnetic field applied along the y-direction, the first node of the Fraunhofer would be at $B_0 = \frac{\phi_0}{(L+2\lambda_L)t} = 390\text{ mT}$ where $L=150\text{ nm}$ is the electrode spacing, $t=20\text{ nm}$ is the thickness of the film and $\lambda_{\text{eff}}=58\text{ nm}$.

In Ref. 34, as the thickness of their 2D electron gas is smaller than our thin film, this interference induced by a field B_y is weaker than in our experiment. In their case, they have to invoke an out-of-plane B_z dipolar magnetic field induced by the in-plane magnetic field to explain the larger decrease of the critical current with the field B_x .

To help address points i and ii, it would be helpful if a full plot of their data from $B=0$ to $B=0.1\text{T}$ is included in the supplementary information.

We added Fig. S7, which shows measurements of the critical current oscillations at two angles from negative ($\sim -100\text{ mT}$) to positive magnetic field ($\sim 100\text{ mT}$), crossing zero magnetic field. As will be seen from these figures, small perturbations near zero magnetic field, likely due to entrance of first vortices, make the noise on the interference pattern large, this is reason why we didn't systematically display the data near zero magnetic field. Despite this larger noise near zero magnetic field, the main Figures Fig.4 and Fig5 as well as Fig. S7 show clearly that the two JIs are in-phase near zero magnetic field. See answer to question below.

iii. It is not clear why the reference device is devoid of any anomalous behavior. The authors state that the two interferometers are in sync at $B=0$ but, due to the nature of the measurement, it is not possible to directly measure that the two are in sync at $B=0$. It is simply inferred from the lowest field at which they show data. Further, the authors state that disorder V_y is absent in the smaller devices, therefore it is possible that the reference device is just that, a device with a sinusoidal current-phase relation and no anomalous value for j . However, looking that the AFM of Fig. S5g, it seems possible that variations of the device parameter are possible on the length scale of 150nm . Therefore, it seems that a convincing argument for the preservations of the important symmetries in the reference device is lacking.!!!

As the referee said, we cannot exclude variations between junctions that could possibly make one junction more disordered than the other. For this reason, the discussion on symmetry is important.

As can be noticed from the symmetry Table I, the presence of disorder V_y would lead to a dephasing at zero magnetic field. The absence of dephasing at zero magnetic field implies that there is no significant disorder V_y that can enable an anomalous dephasing between the two JIs. In the absence of disorder V_y , only a magnetic field B_y can break the last symmetry in table I enabling an anomalous dephasing. Because B_y is null in the reference device, no anomalous dephasing can exist. Thus, our symmetry analysis relies on the absence of dephasing at zero magnetic field.

To make clear that there is no dephasing at zero magnetic field :

We have slightly modified Fig. 5, by adding red and blue dots to highlight the nodes of the oscillations, making obvious that the JIs are in-phase near zero magnetic field and become out-of-phase upon increasing the magnetic field.

We have added Fig. S7. While the JIs shows enhanced noise near zero magnetic field (in particular the reference JI), likely due to the entrance of first vortices, these curves clearly show that the JIs are in-phase at low magnetic field and become out-of phase at higher magnetic field. To make this clear, we added Fig.S7c, showing the dephasing as function of magnetic field, where one can see that the two JIs are in-phase at zero magnetic field and reach a dephasing approaching about $\pi/2$ for an in-plane magnetic field of $\sim 80\text{ mT}$.

We modified a paragraph of the main text to described these additions.

To see this more clearly, the average critical current, shown as a continuous line in Fig.~\ref{Fig5}a, is removed from the critical current curve and the result shown in Fig.~\ref{Fig5}b for the two JIs. On these curves, the nodes at $\pi(2n+1/2)$, $n=0,1,..$, are indicated by large red (blue) dots for the reference (anomalous) curve. At low magnetic field, the two JIs are in-phase as indicated by the blue and red dots being located at the same field position. Upon increasing the in-plane magnetic field, the two JIs become out-of-phase with the anomalous JI oscillating at a higher frequency than the reference JI, as indicated by the blue dot shifting to lower magnetic field position with respect to the red dot. This increased frequency for the anomalous device is expected from Eq.~\ref{JIphase} as a consequence of the anomalous phase shift. Fig.~S7ab shows additional data taken from negative to positive magnetic field, across zero magnetic field. A plot of the phase difference between the two interferometers as function of in-plane magnetic field, shown in Fig.~S7c, demonstrates that the two JIs are in-phase at zero magnetic field and reach a dephasing approaching about $\pi/2$ for an in-plane magnetic field of $\simeq 80$ mT.

iv. The error in extraction of the loop area is large, about two orders of magnitude larger than the value extracted for C_j . It seems possible that this error may be uncoupled to the extraction of C_j , since in the plot of w/w_{ref} only the C_j term is dependent on θ . Yet, I could still see the error in the value of S/S_{ref} coupling into the θ dependence of this curve. Can the authors comment on how the errors S/S_{ref} modify the errors in extracting C_j ?

As described in the main text, using S/S_{ref} and C_j as free parameters, the fit gives $S/S_{ref}=1.023\pm 0.02$, and $C_j=(41\pm 5) 10^{-5}$. Because $C_j \propto \alpha^3$, then : $\frac{\Delta\alpha}{\alpha} = \frac{1}{3} \frac{\Delta C_j}{C_j}$

From the value of C_j , we then obtain : $\alpha = 0.38 \pm 0.015$ eVA.

We add this estimation of the error bar in the main text.

To answer this question about the effect of the error S/S_{ref} on C_j , where $C_j = C_{\varphi_0} \varphi_0 / 2\pi S_{ref}$, we fitted the experimental data by fixing the value S/S_{ref} to different values from $S/S_{ref}=0.98$ to $S/S_{ref}=1.06$. The results are shown in the table and on the Figure below. From the figure, one can see that S/S_{ref} must be within the range $[0.98,1.06]$, beyond this range, the fit deviates too significantly from the data. For $S/S_{ref}=0.98$ (1.06), one finds $C_j=53$ (31). Thus, even with this exaggerate estimation of the error bar, one finds: $C_j = 41 \pm 10 10^{-5}$ which gives $\alpha = 0.38 \pm 0.03$ eVA.

S/S_{ref}	$C_j \times 10^{-5}$
0.98	53 ± 3
1	48 ± 3
1.023	41 ± 3
1.04	36 ± 3
1.06	31 ± 3

1.) How thick is the Bi2Se3 thin films used in this study? The authors should be more clear about where the thin film actually is in the device. I know they attempt this in Fig.S3, but I was still confused about its location.

The thickness of the sample is 20QL, written in the last paragraph before the section result. We modified Fig. S3 by adding a sketch of the lateral view of the device.

2.) What are the areas of the two interferometer? Without this it is not possible for the reader to calculate the extracted anomalous phase from the fit in the plot of $w/w_{ref}(\theta)$.

Thanks for noticing. We added the sentence in the caption of Fig.4:

The reference and anomalous JIs have identical area, respectively S and S_{ref} , where $S \simeq S_{\text{ref}} \simeq 20.6 \mu\text{m}^2$.

Further, can the authors explicitly state the magnitude of the anomalous phase they extract? How does this compare to the anomalous phase predicted for ballistic, diffusive and Dirac systems?

We have added a new table in the main text, providing the experimental dephasing for a magnetic field of 80 mT, compared to the ballistic, diffuse and Dirac contributions. We added the paragraph below in the main text to describe this table.

Table II gives the anomalous phase shift extracted from the critical current oscillations at the largest magnetic field about 80-100 mT. The phase shift is extracted from the magnetic field difference between the last nodes of the oscillations, indicated by blue and red dots on Fig. 5. At this largest magnetic field, we find an anomalous phase shift $\varphi_0 \simeq 0.9\pi$ for all three tilt angles θ . This shows that the anomalous phase shift depends only on the parallel component of the magnetic field as expected. This experimental value is compared with the theoretical values calculated in the ballistic regime, for the Rashba-split conduction states and Dirac states, and in the diffusive regime, for the Rashba-split conduction states. This table shows that the Dirac states provide only a phase shift of 0.005π and so cannot explain the experimental data. The table also shows that the Rashba-split conduction states provide a phase shift of only 0.01π in the ballistic regime while they provide an anomalous shift of 0.94π in the diffusive regime, close to the experimental value, confirming that the junctions are indeed in the diffusive regime and demonstrating the validity of theory leading to Eq. 1.

3.) Which of the devices have 50nm of Al and which have 20nm of Al (the paper states that both are used)? This would be important in understanding the inverse proximity effect, i.e. the effect of the thin Bi₂Se₃ film on the normal state properties of the superconductor (c.f. G. Bergmann PRB 72 134505 (2005)).

The first squid and the device for the microwave measurements have 50nm thick electrodes, the others have 20nm thick electrodes. We have added the thickness of the electrodes in the corresponding captions of the figures.

4.) In the supplementary info, the authors use a sinusoidal variation of the parameter a .

Does the functional form of the variation of a have a significant effect on the resulting Fraunhofer pattern?

No. The functional form only slightly affect the final result. What matter mostly is the length scale over which the parameter α changes. It should be comparable to the junction width.

REVIEWERS' COMMENTS:

Reviewer #1 (Remarks to the Author):

Improvements done by the authors are in my view enough to recommend the manuscript for publication.

Reviewer #2 (Remarks to the Author):

I thank the authors for the response to my questions and for their patience during the wait for my response. I find that their responses do not satiate my curiosity about some of the details of the experiment, as I detail briefly below. However, I feel it unfair to hold this work to a standard uncommonly high for the current field of Josephson junction work in topological materials. Further, I find their results very interesting and believe that the community at large will benefit from the publishing of the results. I only ask that the authors carefully consider the comments as they move forward with their work. Therefore, I believe that this work should be published with minor changes, the execution of which I leave to the authors.

My main concern lies with the origin of the differences between the reference device and the anomalous device. I see that they have added data, but their data is insufficient to show there is no dephasing at zero magnetic field in the reference device. Especially disconcerting is the plot of Fig. S7c, where the solid red lines are haphazardly placed to show the junctions are in phase at $B=0$. Further, the details of the difference of the magnetic-field-dependent oscillations — explained by the authors as a result of flux focusing and vortices penetration — does not make sense. There is a gradual shift in the oscillation period; I would expect the period to experience a discontinuous jump should a vortex enter the junction. Finally, the authors quote a difference in thickness between their device and Ref 34 as a source of discrepancy between the two works. However, given that the thickness is not given in Ref. 34, nor is the thickness of the 2DEG (which could be the entire thickness of the material, or a surface state — both of which are present in Bi₂Se₃ for both nontrivial and trivial reasons) in the present experiment known, it is not possible to make this claim.

I believe that the author should be more open to indicating where they think the data matches their expectations and where it doesn't, leaving open the possibility that other phenomena which are not fully understood might be the source of the difference between the reference and anomalous device. I leave the implementation of this "moral" decision in the hands of the authors.

Moving forward, I would suggest the authors pinpoint the origin of the measured effect, especially the role of nonzero V_y likely to be present in every device they measure. Understanding the origin of these effects may take many from the community, which is why I believe that this paper should still see the light of day.

In red: questions of the referees

We thank both referees and add below the reply to the last comments by the referee 2.

Answers to Reviewer #2:

My main concern lies with the origin of the differences between the reference device and the anomalous device. I see that they have added data, but they data is insufficient to show there is no dephasing at zero magnetic field in the reference device. Especially disconcerting is the plot of Fig. S7c, where the solid red lines are haphazardly placed to show the junctions are in phase at B=0.

The main point of this curve is to show that the anomalous phase shift increases with applied in-plane magnetic field and that there is no anomalous phase shift at zero in-plane magnetic field. Within experimental resolution, both statements are obviously true.

Further, the details of the difference of the magnetic-field-dependent oscillations — explained by the authors as a result of flux focusing and vortices penetration — does not make sense. There is a gradual shift in the oscillation period; I would expect the period to experience a discontinuous jump should a vortex enter the junction.

The major ingredients for explaining the increasing period with in-plane magnetic field are first, the existence of flux-focusing and second, the increasing penetration depth with in-plane magnetic field. The increase of the penetration depth could be due to the penetration of vortices in the electrodes as described in the Supp. Info. In this scenario, the vortices are penetrating the electrodes and do not contribute to the flux in the SQUID. For this reason, they do not lead to jumps in the critical current but only lead to an increase of the penetration depth.

We identified an even simpler scenario. Even in the absence of vortices, the application of an in-plane magnetic field leads to an increase of the penetration depth because of the reduction in the amplitude of the superconducting order parameter at the approach of the upper critical field. According to Tinkham book (page 132, eq. 4.53), the penetration depth increases as $\lambda_{eff} \propto \frac{1}{\Delta(H)} \rightarrow \infty$ for an in-plane magnetic field.

We modified one paragraph in the supplementary info note 3 to clarify this last point.

Finally, the author quote a difference in thickness between their device and Ref 34 as a source of discrepancy between the two works. However, given that the thickness is not given in Ref. 34, nor is the thickness of the 2DEG (which could be the entire thickness of the material, or a surface states — both of which are present in Bi2Se3 for both nontrivial and trivial reasons) in the present experiment known, it is not possible to make this claim.

Regarding the discrepancy between our work and ref. 34, it should be noticed that in our case, the faster reduction of the critical current with **By** than with **Bx** is a phenomena very simple to understand. As described in Supp. Info., **By** leads to additional dephasing in the

junction while not \mathbf{B}_x . The opposite observation of Ref.34 is not obvious to understand. They actually developed a complex scenario to explain it. The origin of this discrepancy is likely to be found in their experiment.

Moving forward, I would suggest the the authors pinpoint the origin of the measured effect, especially the role of nonzero V_y likely to be present in every device they measure. Understanding the origin of these effects may take many from the community, which is why I believe that this paper should still see the light of day.

From the publications of numerous theoretical works predicting the existence of an anomalous phase shift in different contexts (cited in fourth paragraph of manuscript) and the symmetry analysis of Rasmussen2016, it appears that multiple origins for the anomalous phase shift are indeed theoretically possible. For this reason, we have been careful in exploiting symmetry arguments to exclude all other possibilities than the scenario (inplane B_y + finite SOC). While we don't exclude the possibility of disorder along V_y , this disorder does not lead to an anomalous phase shift. For zero in-plane magnetic field, the Fraunhofer pattern of single junctions is always symmetric with respect to B_z and the Josephson interferometers are always in-phase.

This being said, we agree that the theoretical possibility of an anomalous phase shift induced by disorder V_y alone is a quite surprising and interesting prospect. If the required disorder could be designed by microfabrication, this would lead to Josephson circuits with phase-shift at zero magnetic field and zero spin-orbit coupling, a quite remarkable perspective.